# Immune and metabolic markers for identifying and investigating severe Coronavirus disease and Sepsis in children and young people (pSeP/COVID ChYP study): protocol for a prospective cohort study

Sivakumar Oruganti [1], Patrícia R S Rodrigues [2], Daniel White,[2] William John Watkins,[3] Selyf Shapey,[1] Anna Barrow,[1] Rim al Samsam,[1] Sara Ali,[1] Malcolm Gajraj,[1] Richard Skone,[4] Michelle Jardine,[1] Jennifer Evans,[1] Siske Struik,[1] Jong Eun Song,[1] Lloyd Abood,[5] Barbara Paquete,[1] Sian Foulkes,[1] Benjamin Saunders,[6] Angela Strang,[2] Sarah Joanne Kotecha,[7] Bethan Phillips,[8] Awen Evans,[8] Iona Buchanan,[8] Susan Bowes,[8] Begum Ali,[8] Maya Gore,[8] Rhian Thomas-Turner,[8] Robert Andrews,[2] Summia Zaher,[9] Simran Sharma [10,11] Mallinath Chakraborty [12] Edward Parkinson,[2] Federico Liberatore,[2] Thomas Woolley,[2] Sarah Edkins,[2] Luke C Davies,[2] Linda Moet,[2] James E McLaren,[2] Gareth L Watson,[2] Valerie O'Donnell,[13] Kerry Hood,[13] Peter Ghazal [14]

For numbered affiliations see end of article.

**Correspondence to**
Dr Sivakumar Oruganti;
Siva.Oruganti@wales.nhs.uk

## ABSTRACT

**Introduction** Early recognition and appropriate management of paediatric sepsis are known to improve outcomes. A previous system's biology investigation of the systemic immune response in neonates to sepsis identified immune and metabolic markers that showed high accuracy for detecting bacterial infection. Further gene expression markers have also been reported previously in the paediatric age group for discriminating sepsis from control cases. More recently, specific gene signatures were identified to discriminate between COVID-19 and its associated inflammatory sequelae. Through the current prospective cohort study, we aim to evaluate immune and metabolic blood markers which discriminate between sepses (including COVID-19) from other acute illnesses in critically unwell children and young persons, up to 18 years of age.

**Methods and analysis** We describe a prospective cohort study for comparing the immune and metabolic whole-blood markers in patients with sepsis, COVID-19 and other illnesses. Clinical phenotyping and blood culture test results will provide a reference standard to evaluate the performance of blood markers from the research sample analysis. Serial sampling of whole blood (50 µL each) will be collected from children admitted to intensive care and with an acute illness to follow time dependent changes in biomarkers. An integrated lipidomics and RNASeq transcriptomics analyses will be conducted to evaluate immune-metabolic networks that discriminate sepsis and COVID-19 from other acute illnesses. This study received approval for deferred consent.

## STRENGTHS AND LIMITATIONS OF THIS STUDY

⇒ Multimodal investigation of molecular biomarkers for the early diagnosis of paediatric sepsis using a minimally invasive low blood volume.
⇒ Identification of host response molecular patterns associated with various infections (bacterial, viral and fungal).
⇒ Immunophenotyping, immunoassays, lipidomics and functional analysis will be performed during the hospital journey and in cases of suspected sepsis to investigate time-dependent changes in biomarkers of sepsis.
⇒ Potential selection bias towards younger cases.
⇒ Single site observational study.

**Ethics and dissemination** The study has received research ethics committee approval from the Yorkshire and Humber Leeds West Research Ethics Committee 2 (reference 20/YH/0214; IRAS reference 250612). Submission of study results for publication will involve making available all anonymised primary and processed data on public repository sites.

**Trial registration number** NCT04904523.

## INTRODUCTION

Sepsis is a global health problem affecting all age groups. A systematic review of published literature from 1979 to 2016 was carried out to estimate population-based sepsis incidence

in neonates and children. Reports from 15 studies in 12 countries showed an aggregate estimate of 48 (95% CI 27 to 86) sepsis cases and 22 (95% CI 14 to 33) severe sepsis cases in children per 100 000 person-years. The population-level estimate for neonatal sepsis was 2202 (95% CI 1099 to 4360) per 100 000 livebirths, with mortality between 11% and 19%. Extrapolating on a global scale gave an estimated incidence of approximately 3 million cases of sepsis in neonates and 1.2 million cases in children annually. Mortality ranged from 1% to 5% for sepsis and 9% to 20% for severe sepsis.[1]

### Definition of sepsis in children

At present, sepsis is defined in adults as life-threatening organ dysfunction resulting from a dysregulated host response to infection.[2] This definition has removed the terms 'systemic inflammatory response syndrome' (SIRS) and 'severe sepsis' found in previous definitions and highlights organ dysfunction for clinical management. In children, the existing surviving sepsis consensus conference definition of sepsis based on the criteria for SIRS in the presence of confirmed or suspected infection is still used for identification of sepsis and clinical research.[3] The recently updated guidance for the management of sepsis in children (2020) recognises the clinical heterogeneity of this illness in children, accounting for life-threatening organ dysfunction.[4] A revised definition of sepsis by the paediatric sepsis definition task force is awaited.[5] Recently, we put forward a setpoint mechanism for understanding sepsis. This suggests that dysregulation of the host response to infection in sepsis is explainable by regulatory shifts in homeostasis setpoints. This is due to negative and positive feedback changes in the concerted behaviour of multiple (immune, metabolic and neuronal) systems in response to infection. We hypothesise that infants and children have age-dependent metabolic demands and a developing immune setpoint that increases the risk of sepsis.[6] Sepsis can arise from any type of infection—bacterial, viral, fungal or parasitic.[4] Even though less common in children, COVID-19 can be associated with severe illness. History of infection and age-dependent changes in the immune system give rise to subtle but important differences in response to SARS-CoV-2 infection in children when compared with adults. In the early life and younger years, the innate response is a dominant feature of the immune response to infection. In the case of SARS-CoV-2 infection, children are known to mount a stronger and more vigorous interferon (IFN) antiviral response than adults, curtailing viral replication at an early stage of infection.[7]

### Biomarkers in sepsis

Several host inflammatory biomarkers have been suggested to improve the ability for early recognition of sepsis. These include C reactive protein, procalcitonin, interleukin 8, soluble CD14 (presenilin) and triggering receptor expressed on myeloid cells that have all been evaluated without conclusive results.[8] It is important to recognise that sepsis is a complex pathophysiological process comprising an immune response which interacts with the body's metabolic and physiologic functions. A profile of multiple markers capturing the complexity of the pathophysiological response would be better than an individual marker to detect and study the disease process.[9] In neonatal sepsis, it has been shown that the inclusion of metabolic markers as well as adaptive immune responses with innate immune signal changes, increased the accuracy for early recognition of sepsis.[10] These complex molecular changes during sepsis correlate with altered gene expression[11] and in some infectious illnesses, there was a unique signature alteration in gene expression with a demonstrable value in understanding pathogenicity.[12–14] Gene expression markers have been evaluated to identify bacterial infections in febrile children[15] and differences in transcriptomic profiles in blood have been used to discriminate clinical phenotypes and subgroups of sepsis.[16 17]

### COVID-19 in children and young people admitted to paediatric intensive care

Infection with SARS-CoV-2 is mild or asymptomatic in the majority of children.[18 19] The proportion of children presenting to the hospital who have severe COVID-19 and require admission to intensive care has been variable ranging from 1.75% in China[20] to a higher rate of 9.7%–28% in the western world.[21–23] The clinical risk factors for severe COVID-19 in children have been reported to be age over 10 years, black and Asian ethnic groups, comorbidities and obesity.[23 24] The difference in the severity of COVID-19 in children, when compared with adults and the elderly, has been studied extensively. Various factors to explain this difference have been reported and include vasculopathy, the density of ACE 2 receptors, age-based differences in coagulation profiles and immune responses,[25] and heightened antiviral IFN response.[7] Paediatric inflammatory multisystem syndrome temporally related to SARS-CoV-2 (PIMS-TS) also called multisystem inflammatory syndrome in children is a hyperinflammatory syndrome seen in some children following contact with SARS-CoV-2. Two phenotypes have been described—high fevers with severe gastrointestinal symptoms; and shock with myocardial dysfunction and Kawasaki disease-like clinical features. Research to evaluate the pathophysiology of PIMS-TS reported an abnormal immune response mediated by superantigen activity of the SARS-CoV-2 S glycoprotein.[26] It has also been associated with immune markers related to endothelial dysfunction in children.[27] Through this study, we hope to compare immune responses in children with COVID-19, PIMS-TS and sepsis.

### Study methods for molecular and cellular analysis

The studies so far have been restricted to using transcriptomic analysis for an observational case–control methodology to identify differentially expressed genes in children with sepsis (cases) when compared with

children without an acute illness (controls). The evaluation of differentially expressed genes in a particular illness, when compared with healthy controls, helps in understanding the biological pathways and pathophysiology of the disease processes. However, the use of healthy controls in studies aiming to identify gene expression markers that discriminate a particular disease from other illnesses may lead to spectrum bias and fail to account for time-resolved differences.[28] Further, the use of other modal data inclusive of metabolites, in particular lipids, proteomic, cellular and clinical observations allow for a more complete picture of the underlying pathophysiology and diagnostic algorithm development. We propose a prospective cohort methodology to investigate the differentially expressed genes over time in children with sepsis and other acute illnesses.

## METHODS
### Hypothesis and specific aims
A neonatal sepsis classifier integrating immune and metabolic pathways was able to discriminate neonates with bacterial infections and sepsis with high accuracy.[10] We hypothesise that the integrative tripartite immune-metabolic pathways represented by differentially expressed genes between sepsis and other illnesses would discriminate bacterial infection causes of sepsis from COVID-19 illness and other acute non-infectious illnesses. The immune-metabolic pathways vary between different age groups in children under 18 years. Our specific aims include:

► Compare immune and metabolic markers, which discriminate sepsis and COVID-19 from other acute non-infectious illnesses in children admitted to paediatric intensive care unit (PICU).
► Investigate the evolution of immune responses in children with sepsis and COVID-19 using serial samples.
► Evaluate the utility of gene expression markers to discriminate heterogeneous subgroups of sepsis and other critical illnesses with the help of deep clinical phenotyping.
► Understand physiological processes associated with COVID-19 and PIMS-TS and how they differ from bacterial sepsis aetiology.
► Explore age-dependent differences in systemic host response and determine whether these differences are associated with the risk of disease severity to infection.

We plan to explore the feasibility of objectively defining the balance between immune tolerance and resistance in general, as well as in different paediatric age groups. This is through investigating pathway biology response to sepsis and multiomic analyses.

This is a prospective cohort study with a nested case–control analysis. The start date was 30 June 2020 and the proposed end date is 01 June 2023.

### Inclusion criteria
Children or young people under 18 years who:
1. Are admitted to paediatric critical care unit (PCCU).
2. Have an acute illness including trauma.
3. Have routine blood tests as part of their clinical care. (or)

Children or young people under 18 years who are admitted to paediatric wards with confirmed COVID-19 illness or PIMS-TS.

### Exclusion criteria
1. Admitted to hospital for social reasons without an acute illness.
2. Declined consent by parents or carers with legal responsibility, or by competent young person.
3. Admitted to PCCU electively without an acute illness.
4. Direct clinical care team not able to provide research information in a language appropriate for non-English/Welsh speaking participants.

### Patient/family journey
Any child or young person with an acute infectious illness or a non-infectious illness such as trauma on admission to PCCU will be eligible for participation in the study. All children will be screened by the direct clinical care team and research nurse. Research blood samples will be collected in those who are eligible. Data collected will include clinical and laboratory information essential for clinical characterisation including clinical profile, the severity of illness and clinical outcomes. There is no follow-up planned nor repeat blood samples following discharge from the hospital. Anonymous clinical data during the index hospital admission will be collected (figure 1).

### Research blood sample collection
Research blood sample will be collected at admission or onset of an acute illness in eligible children on PICU. In COVID or PIMS-TS patients on paediatric wards and who do not need intensive care, research blood samples will be collected once consent is obtained. Further blood samples may be collected but only alongside clinical blood samples and at least 24 hours apart during the illness, aiming to collect a final sample during recovery from organ failure. All research blood samples will only be collected along with blood samples required for routine clinical care.

### Case definitions and group stratification criteria
We aim to retrospectively carry out deep clinical phenotyping of all patients recruited for the study using anonymous data from the case report forms (CRF) (online supplemental appendix 1). Two clinicians will review the data independently to allocate a diagnosis to all recruited patients. The allocation will be broadly into 'infectious illness', and 'non-infectious illness', with further categorisation into various specific illnesses including COVID-19 and sepsis (online supplemental appendixs 2 and 3—clinical phenotyping tool and clinical phenotyping flow chart, respectively). In case

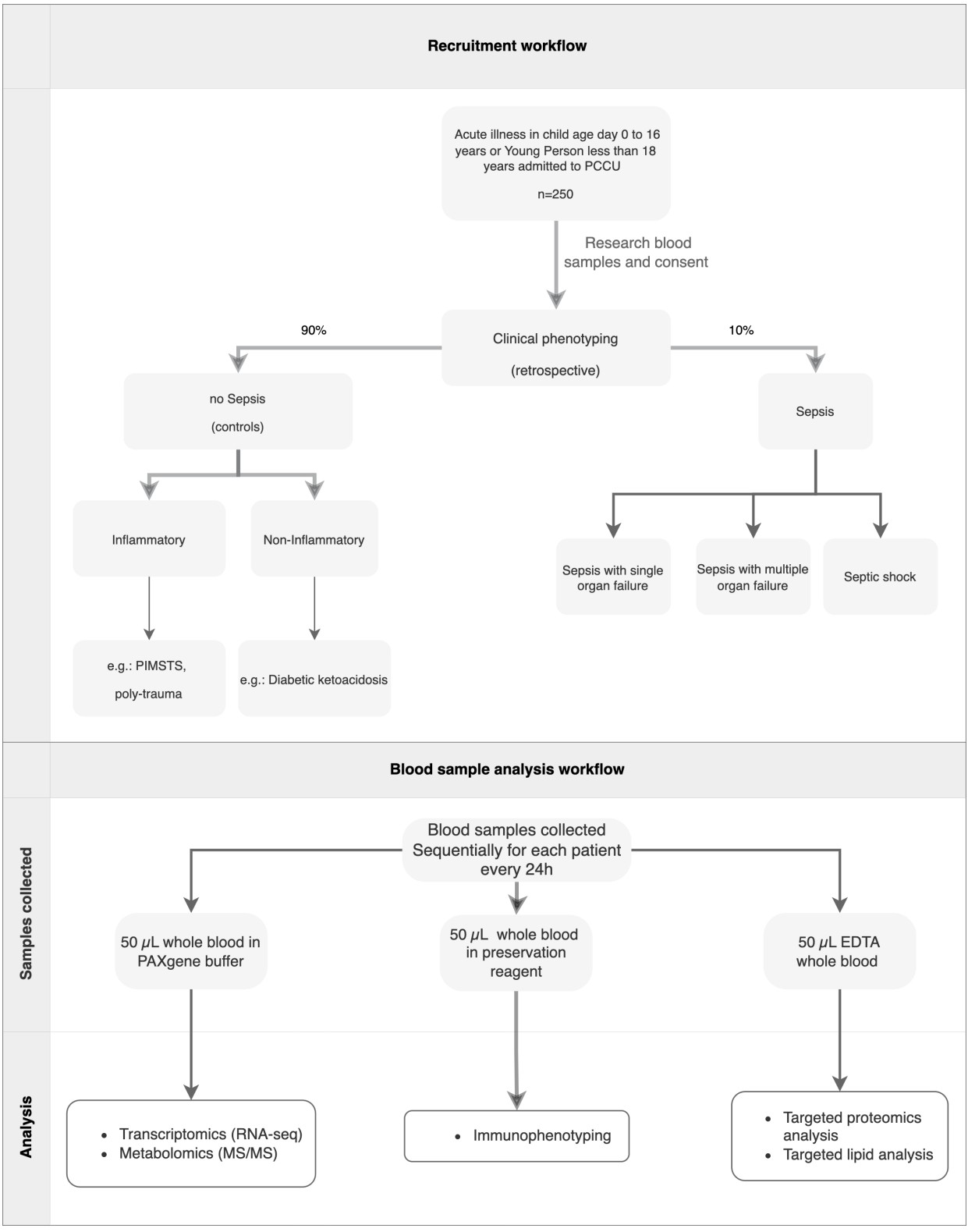

**Figure 1** Patient recruitment and blood sample analysis workflows. EDTA, Ethylene Diamine Tetra acetic Acid MS, mass spectrometry; PAXgene, Potassium Amyl Xanthate gene; PCCU, Paediatric Critical Care Unit; PIMS-TS, Paediatric Multisystem Inflammatory Syndrome Temporally related to SARS-CoV-2.

of any disagreement among the two clinicians, a third clinician will provide a final decision. This forms the reference standard for classification of illness, to aid interpretation of research sample analysis results.

The current practice is to isolate all children admitted to PCCU and wards until their SARS-CoV-2 PCR test result is known. Only those cases that meet the clinical phenotype of COVID-19 illness will be categorised as

COVID-19.[29] Patients who are coincidentally positive for SARS-CoV-2, but identified with other illnesses on clinical phenotyping, will not be categorised as COVID-19.

As the revisions to the definition of sepsis are pending, clinical phenotyping will be broadly based on the surviving sepsis consensus definition (2005). Patients will be included if they meet two or more criteria of SIRS, with at least one criterion related to white cell count or temperature, and with confirmed or strongly suspected to have an infection.[3] There have been algorithms used previously to identify bacterial infections.[15 30] In this study, we plan to evaluate sepsis due to any aetiology including fungal or viral infections. The clinical phenotyping will aid in categorising patients into descriptive clinical clusters, which we believe would have more utility in studying differences in immune-metabolic biological pathways between sepses due to viral, bacterial or fungal sepsis or where there is no microbiological agent identified. The process aims to report those cases as sepsis, which may deviate from the 2005 consensus definition, but have strongly suspected or confirmed infection and life-threatening organ dysfunction as described in the latest surviving sepsis consensus guidance statement.[4] In addition, we can study the differences in sepsis as a whole in comparison with other acute illnesses. In those patients with confirmed infections and who do not meet the SIRS criteria for sepsis, deep clinical phenotyping will help us in evaluating the immune and metabolic markers based on patient outcomes such as severity of illness and organ dysfunction.

### Recruitment and consent
Recruitment of eligible children admitted to PCCU meets the criteria for emergency research as we need to collect research blood samples soon after admission to the critical care unit, along with routine blood tests and it would be inappropriate to approach parents or legal guardians for consent.[31] We have ethics approval for research before consent and collection of multiple samples if the research team find parents not ready to make an informed choice to consent for research. Consultee advice will be sought in patients who are 16 and 17 years old who lack capacity or are not able to make an informed choice due to the critical illness.

In patients admitted to wards with confirmed COVID-19 or PIMS-TS, consent will be obtained after appropriate information sharing before research sample collection at the time of routine clinical blood sampling.

### Patient and public involvement
The study group have engaged with representative members of public for the design of the study. Specific areas where this has led to incorporation of ideas and feedback are listed below:
1. Justification for the Study: Early identification of severe infectious illness will provide a diagnosis and information on prognosis will help in counselling and communicating with patients, families and parents.

2. Representatives of parents have approved of the methods involving research before consent as this is considered research in an emergency setting. The feedback stressed the importance of timing in the approach to parents of prospective research participants, to allow information sharing and consenting in such a way it causes minimum distress. We agreed that the timing for the approach should be based on the parent's and patients' ability to understand the research and not based on the duration from collection of first sample.
3. In view of the importance of appropriate timing for informed consent, it was also agreed that we could take further samples before consent. This was only in a situation where the direct clinical care team and research nurses felt that the parents may be emotionally not ready for approach or prospective research participant was severely unwell.
4. Representative members of the public were involved in drafting information sheets for parents, carers and older children. The final drafts were approved as appropriate for sharing information with parents and patients at the time of initial approach and consenting. Even though feedback about the information provided was incorporated, the information could not be shortened due to regulatory requirements.
5. Feedback from patient groups through Wales Gene Park included the research methods and information shared with families and legal guardians. Also included specific information related to next-generation sequencing in the study. The feedback was supportive of the methods used and suggestions were incorporated into the information sheets.

Peer review and feedback were sought through the UK Paediatric Intensive Care Society—Study Group as well as during a previous grant application.

## SAMPLE PROCESSING AND DATA ANALYSIS
### Sample storage and analysis
The blood samples, at clinical sites, will be stored at −20°C until transferred to the laboratory for processing, where they are stored at −80°C, once consent is obtained (figure 1).

Blood samples transferred to the laboratory will be processed in batches and initially retained by the pSeP team onsite at Cardiff University in a dedicated lock-secured freezer. For long-term storage and access for future research, samples will be housed by the Cardiff Biobank. For the transcriptomic and lipidomic analysis, the blood is mixed with the stabilising reagent potassium amyl xanthate in the collection tubes, all cells are immediately lysed and are not considered to be human tissue. The second tube will be stored as an Ethylene Diamine Tetra acetic Acid (EDTA) whole-blood sample (with or without a stabiliser for cellular phenotyping) and after cell, and targeted proteomics and metabolomics analyses will be banked for future use, for validating future diagnostic platforms.

We aim to apply a systems biology multiomic analysis of blood samples. This will involve using microarray and RNAseq methodology for probing the transcriptome of whole blood. While for specific immune cell characterisation, we will use techniques such as the Cellular Indexing of Transcriptomes and Epitopes by Sequencing (CITE-Seq) methods that perform RNAseq along with quantitative and qualitative information on surface proteins at a single-cell level. For metabolite and proteomic analyses, we will use the methods of liquid chromatography with tandem mass spectrometry (LC-MS/MS) in a targeted high-throughput manner, and we will use quantitative lipidomic profiling (LC-MS/MS) for comprehensive screening of specific pathways such as the complement system.

### Clinical data

CRF will be used for collecting anonymous clinical data, including the clinical classifier (figure 1) from clinical phenotyping (online supplemental appendix 1).

### Sample size calculation for transcriptomic analysis using RNAseq

We aim to recruit 250 patients with acute illnesses admitted to our PICU. Extrapolating the results of previous reports on the prevalence of severe sepsis in the paediatric intensive care population, we estimate that there will be around 25 patients with sepsis included in our study.[32 33] For calculating the power and type 1 error ($\alpha$), we refer to the negative binomial model for biological variability estimations in human samples used by Hart *et al*.[34] Using an RNAseq minimum read depth of 30 million reads per sample and a liberal coefficient of variability ($\sigma$) of 0.6, we calculate that to achieve a Power($\beta$) of 0.8 and type 1 error ($\alpha$) of 0.01, we would need around 20 samples in each group for a twofold change in differentially expressed genes.

### Data processing and analysis

Paired-end reads from RNAseq Illumina sequencing will be trimmed and processed for quality control,[35 36] before mapping to the human genome using Spliced Transcripts Alignment to a Reference.[37] Once we have read counts per gene/transcript, we plan to use R Language and Environment for Statistical Computing[38] followed by quality control and data processing using Bioconductor packages.[39] A per-gene hypothesis of differential average expression will be tested using a negative binomial generalised linear model—DESeq2 package and resulting p values will be controlled for multiple testing using the Benjamini-Hochberg method.[40] For classification, a variety of machine learning and statistical pathway biology approaches, as described in Smith *et al*,[10] will be used. Pathway analyses will be carried out stepwise using a pathway biology approach, becoming more focused. For metabolomic and proteomic data, absolute concentrations (determined by LC-MS/MS) of analytes in extracts from blood samples will apply validated tools such as MetaboAnalyst V.4.0.[41] Further multivariate statistical testing (Principal Component Analysis, Partial Least Squares Discriminant Analysis, Random Forest and Analysis of Variance) using group assignment derived from clinical phenotyping (types of sepsis and between sepsis and non-septic controls) will also be used to determine metabolites or proteins that are significantly different in abundance between clinical phenotypes. Single-cell analyses (CITE-Seq) using surface protein and RNA libraries (10× Genomics) and next-generation sequencing will be multiplexed by cell 'hashing'.[42] Demultiplexed sequencing data will be aligned to the reference transcriptome using CellRanger (10× Genomics) and the number of unique molecular identifiers per cell will be quantified. Computational analyses and quality control will be performed using R packages including Seurat to integrate hashtag, protein and RNA libraries while also enabling demultiplexing of donors, doublet detection and cell clustering.[43]

For the final pathway biomarker assessment of the predictive success of the model, Receiver Operator Characteristics will be applied.

## DISCUSSION

Through this prospective observational cohort study, we plan to evaluate immune-metabolic biological pathways in children with acute illnesses admitted for paediatric critical care. The aim is to evaluate the differences in immune and metabolic pathways in children with sepsis and other illnesses including COVID-19 in children and young people. The use of the cohort design will help in recruiting children of different age groups from birth to under 18 years admitted to PCCU with acute illnesses. The subsequent nested case–control analysis of different clinical groups from the cohort matched for age will help to identify differentially expressed genes among the different clinical phenotypes such as COVID-19 and sepsis with microbiological confirmation, COVID-19 and other viral infections, PIMS-TS and viral infections, PIMS-TS and sepsis with microbiological confirmation, sepsis with microbiological confirmation and confirmed viral infection. The initial analysis in a discovery cohort will be followed by cross-validation in a subsequent validation cohort. Feedback from parents and patient representatives has highlighted the importance of paediatric research for early identification of infections with subsequent poor outcomes or need for organ support. We hope that this study will help in planning future studies addressing this need. We plan to include children with polytrauma as this is considered an example of sterile inflammation.

## ETHICS AND DISSEMINATION

Ethical approval was obtained from the Wales Research Ethics Committee 2 (IRAS project ID250612, REC ref: 20/YH/0214). Operational approval was received from

Health and Care Research Wales and local resource and capacity were supported by the research and development department of the Cardiff and Vale University Health Board. The study is sponsored by Cardiff University. The sponsor or funder does not have any role in the study design, data collection or analysis. Submission of study results for publication will involve making available all anonymised primary and processed data on public repository sites. All study results will be presented at national and international conferences and published in peer-reviewed open-access journals.

**Author affiliations**
¹Paediatric Critical Care Unit, Noah's Ark Children's Hospital for Wales, Cardiff, UK
²School of Medicine, Cardiff University, Cardiff, UK
³Cochrane Institute of Primary Care and Public Health, Cardiff University, Cardiff, UK
⁴Department of Paediatric Intensive Care, Noah's Ark Children's Hospital for Wales, Cardiff, UK
⁵Morriston Hospital, Swansea, UK
⁶Infectious Diseases services for Wales, Noah's Ark Children's Hospital for Wales, Cardiff, UK
⁷Department of Child Health, Cardiff University, Cardiff, UK
⁸Children's and Young Adults Research Unit, University Hospital of Wales, Cardiff, UK
⁹Department of Obstetrics and Gynaecology, University Hospital of Wales, Cardiff, UK
¹⁰Infection and Immunity, Cardiff University, Cardiff, UK
¹¹Women's Unit, Cardiff and Vale NHS Trust, Cardiff, UK
¹²University Hospital of Wales, Cardiff, UK
¹³Centre for Trials Research, College of Biomedical and Life Sciences, Cardiff University, Cardiff, UK
¹⁴Medical School, Cardiff University, Cardiff, UK

**Acknowledgements** The authors would like to acknowledge the following contributions: Helen Thomas, Benjamin Beach and Sarra Beach (representing parents and members of public) contributed to consent methods and information for patients, families and parents. Representative members of public in Project Sepsis and Wales Gene Park contributed feedback on consent methods and information for patients, families and parents; Professor Mark Peters (consultant paediatric intensivist, Great Ormond Street Hospital): Through the UK Paediatric Intensive Care Society Study Group, Professor Peters provided feedback on study design.

**Contributors** SO (principal investigator): Conceptualised the study and contributed to design, engagement with representative members of public, sample size calculation, recruitment of patients, data acquisition, analysis, interpretation of results, writing and editing the manuscript. PRSR: Investigation, methodology, visualisation. DW: Investigation, methodology. WJW: Contributed to sample size considerations and data analysis. SShapey, AB, RaS, SA, MGajraj, RS, MJ, JE, SStruik, JES, LA, BPaquete, SF and BS were involved in the recruitment of patients, data acquisition, clinical phenotyping, analysis and writing of the manuscript. AS: Ethics, resources, data curation, project administration (project manager). SJK: Governance and study administration. BPhillips, AE, IB, SB, BA, MGore and RT-T contributed to drafting data collection forms, sample storage methods, patient recruitment, data collection and governance. RA: Formal analysis. SZ and SSharma: Contributed to patient information and consenting. MC: Contributed to considerations around methodology, consenting, sample collection procedures and editing the manuscript. EP and FL: Quality assurance, quality control and machine learning analyses. TW: Mathematical modelling and testing setpoint hypothesis. LCD: Investigation, methodology. SE: Resources, investigation, project administration, methodology. LM: Investigation, methodology. JEML: Investigation, methodology. GLW and KH: Electronic case report form and data management. VO'D: Methodology, resources. PG (chief investigator) contributed to conceptualisation, design, engagement with representative members of public, sample analysis methods, data analysis, interpretation of results, writing and editing of the manuscript.

**Funding** The study has received funding from the Welsh Government-EU ERDF for Ser Cymru II programme grant (East Wales ERDF Programme grant number 80762-CU-106) for Project Sepsis (PG). PRSR was supported in part by MRC (R520481) to PG.

**Competing interests** None declared.

**Patient and public involvement** Patients and/or the public were involved in the design, or conduct, or reporting, or dissemination plans of this research. Refer to the Methods section for further details.

**Patient consent for publication** Not applicable.

**Provenance and peer review** Not commissioned; externally peer reviewed.

**ORCID iDs**
Sivakumar Oruganti http://orcid.org/0000-0003-1384-2664
Patrícia R S Rodrigues http://orcid.org/0000-0003-0768-0013
Simran Sharma http://orcid.org/0000-0002-6647-9355
Mallinath Chakraborty http://orcid.org/0000-0002-1721-6532
Peter Ghazal http://orcid.org/0000-0003-0035-2228

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
