## [Reviewer comments · BMJ Open]

ARTICLE DETAILS

TITLE (PROVISIONAL)	Immune and metabolic markers for identifying and investigating severe Coronavirus disease and Sepsis in children and young people. (pSeP/COVID ChYP study) – Protocol for a prospective cohort study
AUTHORS	Oruganti, Sivakumar; Rodrigues, Patrícia R.S.; White, Daniel; Watkins, W John; Shapey, Selyf; Barrow, Anna; al Samsam, Rim; Ali, Sara; Gajraj, Malcolm; Skone, Richard; Jardine, Michelle; Evans, Jennifer; Struik, Siske; Song, Jong; Abood, Lloyd; Paquete, Barbara; Foulkes, Sian; Saunders, Benjamin; Strang, Angela; Kotecha, Sarah; Phillips, Bethan; Evans, Awen; Buchanan, Iona; Bowes, Susan; Ali, Begum; Gore, Maya; Thomas-Turner, Rhian; Andrews, Robert; Zaher, Summia; Sharma, Simran; Chakraborty, Mallinath; Parkinson, Edward; Liberatore, Federico; Woolley, Thomas; Edkins, Sarah; Davies, Luke C.; Moet, Linda; McLaren, James E.; Watson, Gareth L.; O'Donnell, Valerie; Hood, Kerry; Ghazal, Peter

VERSION 1 – REVIEW

REVIEWER	Daniela Medeiros Hospital Israelita Albert Einstein São Paulo
REVIEW RETURNED	05-Dec-2022

GENERAL COMMENTS	Thanks for the opportunity to review this manuscript. The suggestions are below: 1) Pg 7 / Line 34: to include patients with sepsis2) Pg 30 / Line 36: to include meropenem as an antibiotic3) Pg 31 / Line 37: to include micafungine as an antifungal4) The routine of blood sample analysis it is not clear on the main text. The "blood analysis workflow" is not cited on the main text. I think it is important to describe the if there is a time of daily blood sample collection. The suggestion is to include an item called "blood sample routine collection" or something similar. As a reader, I have the understanding that there is no daily blood collection. (Pg 7/ lines 56-58). Blood collection in children is an important point for parents and for study adhesion.
--

REVIEWER	Alexandros Rovas University of Münster
REVIEW RETURNED	08-Feb-2023

GENERAL COMMENTS	I have read with interest the protocol for the study entitled “ Immune and metabolic markers for identifying and investigating severe Coronavirus disease and Sepsis in children and young people. (pSeP/COVID ChYP study)”. I have some comments I would like to make:
---

	 - The inclusion criteria should be defined in a more accurate way. In the recruitment workflow the presence or absence of sepsis is defined by its “clinical phenotype”. The authors should define, if the categorization will be in a retrospective or prospective way (on admission) and if it will be based on SIRS criteria, sepsis 2 or sepsis 3 definition. At the same time, in the sepsis subcategories the sepsis 3 definition (organ failure) is used. - In the workflow, COVID-19 belongs to “no sepsis” category. However, COVID-19 is a viral sepsis. At the same time there is another workflow in the CRF, where the categorization is different. - Will patients with positive COVID-19 swab test included even if that is not the main reason of admission? - How will patients that end up having an infection without fulfilling sepsis criteria will be treated regarding categorisation? - In the text it states that “no follow up is planned”, however in the CRF is referred to a follow up letter. - How will the follow-up during the hospital stay take place? Are the timepoints predefined?
--	---

VERSION 1 – AUTHOR RESPONSE

Reviewer: 1

Dr. Daniela Medeiros, Hospital Israelita Albert Einstein São Paulo Comments to the Author:

Thanks for the opportunity to review this manuscript. The suggestions are below:

1) Pg 7 / Line 34: to include patients with sepsis.

Author response: We would like to thank Dr Daniela Medeiros for the valuable feedback. We have reviewed this and would like to clarify that all children admitted to the paediatric intensive care unit with an acute illness would be included in the study. Of the children admitted to other paediatric wards, only those with COVID-19 or PIMS-TS would be included.

2) Pg 30 / Line 36: to include meropenem as an antibiotic

Author response: Meropenem is already cited in the list of antibiotics. Even though the list is not exhaustive, we have included an ‘other’ option to capture any other antibiotics.

3) Pg 31 / Line 37: to include micafungine as an antifungic

Author response: Thank you for the suggestion to include micafungine. We did not include this anti-fungal agent in the list as it is not commonly used in our centre. However, there is an ‘other’ option to collect data on any other anti-fungal agents used.

4) The routine of blood sample analysis it is not clear on the main text. The "blood analysis workflow" is not citted on the main text. I think it is important to describe the if there is a time of daily blood sample collection. The suggestion is to include an item called " blood sample routine collection" or something similar.

As a reader, I have the understanding that there is no daily blood collection. (Pg 7/ lines 56-58). Blood collection in children is an important point for parents and for study adhesion.

Author response: Thank you for highlighting the lack of clarity about research blood sample collection. We have addressed this by including a separate section about research blood sample collection.

Reviewer: 2

Dr. Alexandros Rovas , University of Münster Comments to the Author:

I have read with interest the protocol for the study entitled “ Immune and metabolic markers for identifying and investigating severe Coronavirus disease and Sepsis in children and young people. (pSeP/COVID ChYP study)”.

I have some comments I would like to make:

- The inclusion criteria should be defined in a more accurate way. In the recruitment workflow the presence or absence of sepsis is defined by its “clinical phenotype”. The authors should define, if the categorization will be in a retrospective or prospective way (on admission) and if it will be based on SIRS criteria, sepsis 2 or sepsis 3 definition. At the same time, in the sepsis subcategories the sepsis 3 definition (organ failure) is used.

Author response: Thank you for the comments. We aim to include all children with acute illness in paediatric intensive care unit. Hence the inclusion criteria may seem non-specific. We agree that it is not clear whether the categorization based on clinical phenotype is retrospective or prospective. It is done retrospectively, by clinicians not directly involved in care of the included patient. We have now included this detail in the manuscript. (page8, line9)

The clinical phenotyping for sepsis is based on Systemic Inflammatory Response Syndrome (SIRS) criteria. We aim to include under sepsis, those who may deviate from the SIRS criteria but have organ dysfunction associated with probable or confirmed infections (sepsis associated organ dysfunction), as described in the recent Surviving Sepsis consensus conference statement published in 2020 [1]. We have amended the manuscript to make it clearer.

- In the workflow, COVID-19 belongs to “no sepsis” category. However, COVID-19 is a viral sepsis. At the same time there is another workflow in the CRF, where the categorization is different.

Author response: We would like to thank Dr Alexandros Rovas for pointing out this discrepancy. We agree that COVID-19 can be a form of viral sepsis and this should not have been included under the ‘non septic inflammatory illnesses’ in Figure 1 workflow. We have now amended this in the manuscript.

- Will patients with positive COVID-19 swab test included even if that is not the main reason of admission?

Author response: We would only include those children with SARS-CoV2 infection, who have the manifestations of COVID illness.

- How will patients that end up having an infection without fulfilling sepsis criteria will be treated regarding categorisation?

Author response: We would like to thank Dr Rovas for raising an important issue in this line of work. We expect that there may be a group of patients who have confirmed microbiological evidence of infection but not meet criteria of sepsis. We would not be classifying them as sepsis. However, this study would give us the opportunity to evaluate biological markers and immune-metabolic response pathway differences in this group of patients and those classified as sepsis. In those patients with confirmed infections, deep clinical phenotyping will help us in evaluating the immune and metabolic markers based on patient outcomes such as severity of illness and organ dysfunction. We hope that this will help in planning future studies focussed on identifying those patients with infections at risk of organ dysfunction and poor patient outcomes. Feedback from parents and patient representatives as well as a recent scoping review by one of the authors have highlighted the identification of infections with subsequent poor outcomes or need for organ support, as an important topic for research in children [2].

- In the text it states that “no follow up is planned”, however in the CRF is referred to a follow up letter.

Author response: There is no follow up planned for patients included in the study. The follow up mentioned in the CRF is for obtaining postal consent in those patients admitted to paediatric intensive care unit and who are not consented during stay in hospital. This study has been approved for deferred consent as it meets the criteria for emergency research. Only those patients with consent will be included in the study.

- How will the follow-up during the hospital stay take place? Are the timepoints predefined?

Author response: There are no pre-defined time points. As research samples are collected only at the time of routine blood tests for clinical management, it is not possible to have pre-specified time points. However, in those patients on organ support in paediatric intensive care, we envisage at least daily routine blood tests as part of normal clinical management. Hence in such situations, we hope to have research blood samples taken at least every 24 hours.